# Prevalence and Outcomes of Acute Hypoxaemic Respiratory Failure in Wales: The PANDORA-WALES Study

**DOI:** 10.3390/jcm9113521

**Published:** 2020-10-31

**Authors:** Maja Kopczynska, Ben Sharif, Richard Pugh, Igor Otahal, Peter Havalda, Wojciech Groblewski, Ceri Lynch, David George, Jayne Sutherland, Manish Pandey, Phillippa Jones, Maxene Murdoch, Adam Hatalyak, Rhidian Jones, Robert M. Kacmarek, Jesús Villar, Tamas Szakmany

**Affiliations:** 1Department of Anaesthesia, Intensive Care and Pain Medicine, Division of Population Medicine, Heath Park Campus, Cardiff University, Cardiff CF14 4XN, UK; maya.kopczynska@gmail.com (M.K.); SharifBS@cardiff.ac.uk (B.S.); 2Salford Royal NHS Trust, Stott Lane, Manchester M6 8HD, UK; 3Anaesthetic Department, Royal Glamorgan Hospital, Cwm Taf Morgannwg University Health Board, Llantrisant CF72 8XR, UK; Ceri.Lynch5@wales.nhs.uk; 4Anaesthetic Department, Glan Clwyd Hospital, Betsi Cadwaladr University Health Board, Bodelwyddan, Rhyl LL18 5UJ, UK; Richard.Pugh@wales.nhs.uk; 5Anaesthetic Department, Glangwili Hospital, Hywel Dda University Health Board, Carmarthen SA31 2AF, UK; Igor.Otahal@wales.nhs.uk (I.O.); Peter.Havalda@wales.nhs.uk (P.H.); 6Anaesthetic Department, Withybush Hospital, Hywel Dda University Health Board, Haverfordwest SA61 2PZ, UK; Wojciech.Groblewski@wales.nhs.uk; 7Anaesthetic Department, Wrexham Maelor Hospital, Betsi Cadwaladr University Health Board, Wrexham LL13 7TD, UK; David.George@doctors.org.uk; 8Ed Major Critical Care Unit, Morriston Hospital, Swansea Bay, University Health Board, Swansea SA6 6NL, UK; Jayne.Sutherland@doctors.org.uk; 9Critical Care Department, University Hospital Wales, Cardiff and Vale University Health Board, Cardiff CF14 4XW, UK; manish55pandey@gmail.com; 10Critical Care Directorate, Royal Gwent Hospital, Aneurin Bevan University Health Board, Newport, Gwent NP20 2UB, UK; phillippajones@gmail.com (P.J.); Maxene.Murdoch@doctors.org.uk (M.M.); 11Critical Care Directorate, Nevill Hall Hospital, Aneurin Bevan University Health Board, Abergavenny NP7 7EG, UK; adam.hatalyak@gmail.com; 12Anaesthetic Department, Princess of Wales Hospital, Cwm Taf Morgannwg University Health Board, Bridgend CF31 1RQ, UK; Rhidianjones@hotmail.com; 13Department of Respiratory Care, Massachusetts General Hospital, Boston, MA 02114, USA; rkacmarek@partners.org; 14Department of Anesthesia, Harvard University, Boston, MA 02115, USA; 15CIBER de Enfermedades Respiratorias, Instituto de Salud Carlos III, 28029 Madrid, Spain; jesus.villar54@gmail.com; 16Research Unit, Hospital Universitario Dr. Negrín, 35010 Las Palmas de Gran Canaria, Spain

**Keywords:** respiratory failure, ventilation, rescue therapies, survival

## Abstract

Background: We aimed to identify the prevalence of acute hypoxaemic respiratory failure (AHRF) in the intensive care unit (ICU) and its associated mortality. The secondary aim was to describe ventilatory management as well as the use of rescue therapies. Methods: Multi-centre prospective study in nine hospitals in Wales, UK, over 2-month periods. All patients admitted to an ICU were screened for AHRF and followed-up until discharge from the ICU. Data were collected from patient charts on patient demographics, clinical characteristics, management and outcomes. Results: Out of 2215 critical care admissions, 886 patients received mechanical ventilation. A total of 197 patients met inclusion criteria and were recruited. Seventy (35.5%) were non-survivors. Non-survivors were significantly older, had higher SOFA scores and received more vasopressor support than survivors. Twenty-five (12.7%) patients who fulfilled the Berlin definition of acute respiratory distress syndrome (ARDS) during the ICU stay without impact on overall survival. Rescue therapies were rarely used. Analysis of ventilation showed that median Vt was 7.1 mL/kg PBW (IQR 5.9–9.1) and 21.3% of patients had optimal ventilation during their ICU stay. Conclusions: One in four mechanically ventilated patients have AHRF. Despite advances of care and better, but not optimal, utilisation of low tidal volume ventilation, mortality remains high.

## 1. Introduction

Worldwide over 100 million patients are ventilated annually, mostly in operating theatres, with about 10 million ventilated in intensive care units (ICU). It is estimated that approximately one million patients develop acute hypoxemic respiratory failure (AHRF), although data on the exact prevalence and outcomes of this condition are sparse [1,2]. Acute respiratory distress syndrome (ARDS) is an acute inflammatory lung injury associated with AHRF [3]. Currently, the diagnosis of ARDS is challenging and relies on patient characteristics, such as a history of a predisposing illness, an acute onset, as well as radiological and physiological measurements [4]. Despite the introduction of a new ARDS definition by the Berlin criteria, it is still associated with mortality greater than 50% [3,5]. Although there are some previously published observational studies examining the incidence and mortality of patients with AHRF and ARDS, there are no studies specifically assessing the epidemiological characteristics, patterns of ventilation and clinical outcomes in patients with acute hypoxemic respiratory failure in the current era of lung protective ventilation [2,6,7]. Understanding the factors associated with AHRF outcomes could lead to development of effective interventions and improvements in care.

The primary aim of our “Prevalence and Outcome of Acute Hypoxemic Respiratory Failure in Wales (PANDORA-WALES)” study was to identify the prevalence of AHRF in ICU in Wales as well as its associated mortality. The secondary aim of the study was to investigate the values of parameters of ventilatory management as well as the use of rescue therapies for hypoxemia, such as prone positioning, recruitment manoeuvres and extracorporeal assist.

## 2. Experimental Section

### 2.1. Study Design and Participants

This was a multi-centre prospective observational study carried out in nine hospitals in Wales, United Kingdom. Patients were enrolled during two periods of two consecutive months (1 October 2017 to 30 November 2017 and 1 February 2018 to 31 March 2018). Data collection was performed using a secure digital data collecting platform [8].

During the study, consecutive patients admitted to ICU were screened daily and recruited to the study if they fulfilled all inclusion criteria: age ≥ 18-years old; endotracheal intubation and mechanical ventilation; PaO_2_/FiO_2_ ≤ 300 mmHg on invasive mechanical ventilation with a PEEP of 5 cm H_2_O or more, and with a FiO_2_ of 0.3 or more. Data were collected on patient demographics: reason for ICU admission, methods for diagnosis of the cause of AHRF, physiological and laboratory results, treating team management and complications during the ICU stay. Patients were followed-up until discharge from the ICU.

Data management, monitoring and reporting of the study were performed in accordance with the Good Clinical Practice Guidelines. The study was registered with each participating University Health Board and internal risk reviews classed it as a service evaluation, with no formal ethical approval required. The study was prospectively registered at an international registry (NCT03358043).

### 2.2. Statistical Analysis

Categorical variables are described as proportions and are compared using a Chi-square test. Continuous variables are described as median and inter-quartile range and compared using a Mann-Whitney U test between the groups. We compared patient groups ventilated with different modalities and survivors vs. non-survivors. We used univariate regression analysis to determine risk factors of mortality. A two-tailed *p*-value < 0.05 was considered statistically significant. All statistical tests were calculated using SPSS 23.0 (SPSS Inc., Chicago, IL, USA).

## 3. Results

### 3.1. Patient Characteristics

There were 2215 critical care admissions in the participating hospitals during the study period, with 886 patients receiving mechanical ventilation. A total of 197 (22.2%) patients met all of the inclusion criteria of AHRF and were recruited to the study (Figure 1). Median patient age was 60 years (IQR 49–70) and 59.4% (117/197) of patients were males. The summary of patient demographics and clinical characteristics is summarized in Table 1.

The diagnosis of the cause of respiratory failure was performed using chest X-rays in the majority of the patients (86.3%, 170/197). Other modalities of diagnosis included CT scan (24.4%, 48/197); blood culture (19.8%, 39/197); sputum culture (18.8%, 37/197); echocardiography (14.2%, 28/197); bronchoscopy (8.6%, 17/197) and bronchoalveolar lavage (4.1%, 8/197). Lung biopsy and autopsy post-mortem were not performed in any of the cases.

The causes of AHRF were as follows: pneumonia (29.9%, 59/197), sepsis (20.8%, 41/197), cardiogenic shock (12.7%, 25/197), central nervous system disorders (8.1%, 16/197), chronic obstructive pulmonary disease exacerbation (5.6%, 11/197), hypovolaemic shock (4.6%, 9/197), drug overdose (3.6%, 7/197), asthma exacerbation (2.5%, 5/197), chest trauma (2.5%, 5/197), pancreatitis (2%, 4/197), pulmonary oedema (2%, 4/197), malignancy (1%, 2/197), pneumothorax (1%, 2/197), lung empyema (1%, 2/197), pulmonary embolism (1%, 2/197) and other (1.5%, 3/197).

### 3.2. Patient Ventilation

Analysis of all the measurements during the whole ICU stay showed that median V_T_ was 7.1 mL/kg PBW (IQR 5.9–9.1) (Figure 2A). PEEP of <6 cm H_2_O was used in 43.9% patients, 6–10 cm H_2_O in 45.8%, 11–15 cm H_2_O in 8.8% and over 15 cm H_2_O in 1.5% (Figure 2B).

We also investigated what proportion of patients received an optimal V_T_ (less than 6 and between 6 and 7 mL/kg PBW) by analysing daily V_Tmax_ and V_Tmin_ during the whole ICU stay. A total of 21.1% of all V_Tmax_ measurements were within the optimal range and 75.5% of V_Tmin_ measurement were optimal. For V_Tmax_, 77.9% of measurements were above 7 mL/kg PBW and 7.8% were below 6 mL/kg PBW. For V_Tmin_, 23.7% of measurements were above 7 mL/kg PBW and 47.5% were below 6 mL/kg PBW. Forty-two patients (21.3%) had an optimal ventilation at some point during the ICU stay.

The majority of patients were treated with volume controlled (64%, 126/197) and pressure controlled (28.9%, 57/197) ventilation. Five patients were treated with other ventilation types: one with assisted spontaneous breathing and four with continuous positive airway pressure. We have investigated an impact on ventilation type on delivered V_T_ values (Table 2). Volume controlled ventilation was associated with higher proportions of optimal V_T_ (*p* = 0.049) and lower proportions of V_T_ above 7mL/kg PBW (*p* = 0.038).

### 3.3. Patient Management

Prior to enrolment in the study and meeting respiratory failure criteria, 9.6% patients (19/197) were treated with non-invasive ventilation and 19.8% (39/197) received high flow nasal oxygen therapy. On the day of recruitment and throughout the ICU stay, patients received other interventions and monitoring modalities presented in Table 3. On the day of recruitment, one in five patients were treated with continuous neuromuscular blockade and one in six had recruitment manoeuvres performed. Prone positioning was rarely used either as an initial treatment option or as a rescue therapy.

### 3.4. Complications and Survival Analysis

A significant proportion of patients suffered from complications, both respiratory and systemic such as shock or sepsis as presented in Table 4.

Out of 197 recruited patients, 70 (35.5%) were non-survivors, 125 (63.5%) were survivors and data were missing for 2 (1.0%). The univariate analysis of factors associated with mortality is presented in Table 5.

Twenty-five (12.7%) patients fulfilled the Berlin definition of ARDS at any time during the ICU stay. Median number of days between ICU admission and ARDS development was 0 (IQR 0–1). Out of all patients with ARDS, 5 (36%) were non-survivors. Development of ARDS did not have an impact on overall survival in our patient cohort (*p* = 0.37).

## 4. Discussion

This is the first study prospectively investigating the prevalence, management processes and outcomes of patients developing AHRF in ICUs in Wales. A total of 22.2% of patients admitted for mechanical ventilation developed AHRF, which was associated with a significant mortality rate of 35.5%, similar to other European studies [1,9].

AHRF appears to be common in the ICUs [10,11], affecting almost a quarter of those needing mechanical ventilation. However, most large-scale prevalence studies concentrate on either patients undergoing mechanical ventilation or those meeting defined criteria for acute lung injury/ARDS [9,12]. In the multi-national LUNG-SAFE study, 34% of ventilated patients fulfilled AHRF criteria, but with no data showing if this differs across countries [13]. It must be noted that the LUNG-SAFE study had a significant bias towards tertiary centres, whilst our study was mainly conducted in district general hospitals, which could lead to a different case-mix and could explain our lower prevalence [14]. We found that the incidence of ARDS using the Berlin definition [3] was almost five times lower than in LUNG-SAFE study conducted three years earlier—12.7% vs. 67.2% of patients with AHRF, respectively [13]. This observed discrepancy highlights the difficulties in adopting the Berlin definition as inclusion criteria for clinical trials [15]. Previous studies indicated that the PaO_2_/FiO_2_ ratio-based severity grading is open for interpretation and that measurements taken at a standardised ventilator setting may change the severity grading of more than half of ARDS patients [4]. Importantly, in the LUNG-SAFE study approximately 40% of patients diagnosed with ARDS, were not deemed to have the condition by the treating clinician [13]. Based on our data it is difficult to ascertain whether ARDS was “underdiagnosed” as postulated by the LUNG-SAFE study, or whether the clinicians interpreted the chest infiltrates appropriately in our study and the incidence of ARDS in Wales is within the previously described 10–15% category [9,13]. ARDS was prevalent in ICU admission in our cohort and less likely to develop as a complication of mechanical ventilation, in line with recent large international studies [10,16].

The outcome of AHRF in our study was similar to what was reported in the same period in several cohorts of mechanically ventilated hypoxaemic patients [10,17,18]. However, the observed mortality was almost 10% higher compared to cohorts in Italy and in Canada with similar demographics and similar degrees of hypoxaemia [19,20]. There are several possible explanations for these discrepancies. There is a well-documented disparity of critical care beds in the UK and in particular in Wales, compared with other developed countries [21,22]. This, in general, leads to higher acuity admissions to the ICU, with higher underlying risk of death, which has been described previously [23]. Indeed, our patient population was older, was more likely to have multi-organ failure based on the SOFA score and had higher prevalence of underlying medical problems compared to other studies [9]. Unsurprisingly, these previously well-described risk factors were associated with mortality in our cohort [9,24]. However, there is also a possibility that care processes and adherence to best practice guidance in mechanical ventilation was better in the studies conducted in Italy and Canada, accounting for better mortality [19,20]. In our cohort of patients with AHRF only one in five received optimal ventilation during their ICU stay, recommended as <7 mL/kg PBW [25]. Four out of five patients had V_Tmax_ values above 7 mL/kg PBW and almost a quarter of the measurements of V_Tmin_ were outside the recommended range. This finding is consistent with other studies in which patients received larger tidal volumes than expected [16]. We opted to use a strict criterion for optimal V_T_ as it has been known for decades that the physiological V_T_ is 6 mL/kg PBW in humans [26]. It has been shown that ventilation with use of large tidal volumes can cause hypoxaemia and release of inflammatory mediators, leading to increase in lung inflammation and injury to other organs [27]. Several retrospective analyses of large observational studies suggested that every 1mL/kg PBW increase of V_T_ is also associated with higher mortality [28,29]. Although in the recent PRoVENT multi-centre prospective study of mechanically ventilated patients at risk of ARDS, Neto et al. reported that V_T_ was not different between survivors and non-survivors, the patient population was significantly different compared to ours, as their patients had a much higher PaO2/FiO2 ratio and were more likely to be admitted following surgical procedures [12]. Importantly, the median V_T_ in our observations was 7 compared to 8 mL/kg PBW in the PRoVENT study, continuing the trend of V_T_ reduction over time in various trials [9,11,12,13]. Whilst a multi-centre randomized controlled trial (RCT) did not find difference in ventilator free days or mortality when comparing low to intermediate V_T_ strategies in patients without ARDS, the PReVENT trial had inadequate separation between the two arms after day 1, which means this study was likely underpowered to detect any true difference [30]. Our findings are in-line with the outcomes of the low V_T_ arm of the PReVENT trial, which has targeted 7 mL/kg PBW [30].

Rescue therapies were not used in the vast majority of the patients. Neuromuscular blocking agents were used only for 21.3% and recruitment manoeuvres for 17.8% of patients on the day of diagnosis of AHRF, whereas prone positioning and more expensive and invasive technologies such as ECMO were used in less than 1% of patients, which is significantly lower than in the study by Bellani et al. [13]. At the time of the study, both neuromuscular blockade and prone positioning was recommended for severe ARDS based on previously published RCT, whilst uncertainty existed around the use of recruitment manoeuvres [5,31,32]. However, as the majority of our patients did not meet ARDS criteria, it is possible that the relatively low use of adjunctive measures reflects ongoing uncertainty about the quality of evidence supporting these interventions. Indeed, recent RCTs put some doubt about the efficacy of these interventions in the moderate to severe hypoxaemia category, with individualised ventilation strategies, including prone positioning, shown to be harmful compared to standard care with a low PEEP strategy [33]. Importantly, the harm measured in the experimental group was mostly attributed to the misclassification of focal or diffuse changes on chest imaging, further emphasising the difficulty of ARDS diagnosis using these modalities [33]. At the beginning of the study, the ART trial was published, reporting increased harm from a recruitment manoeuvre strategy in moderate to severe ARDS, where patients had similar levels of physiological derangement to our population [34]. This publication, which has been widely discussed in conventional and social media, might have had an impact on the use of recruitment manoeuvres during our data collection period. More recently, the use of early neuromuscular blockade has been shown to be inferior to the early mobilisation, awakening and spontaneous breathing trial approach in a large, multi-centre study recruiting patients at the severe end of the hypoxaemic respiratory failure spectrum [35]. These results cast significant doubt on the efficacy and safety of deep muscle paralysis in AHRF and our findings regarding neuromuscular blockade use are in line with the standard care arm of the ROSE-PETAL study [35].

The strengths of the study include wide participation of centres in Wales including both academic centres and general district hospitals. Capturing data over a four-month period was implemented to ensure representativeness. Information was collected prospectively enabling the analysis of real-life use of ventilation and rescue therapies. A similar study, using the same dataset, has been conducted in Spain (NCT03145974). This shows that this type of assessment of ICU performance could be standardized and used internationally. 

There are significant limitations to our study: although data collection was digitised, it was still labour intensive, requiring everyday screening, meaning there is the potential that some patients were missed and not recruited in the study, however, retrospective reconciliation with ICU admissions did not reveal any. We have concentrated our data collection on the winter months, when AHRF incidence could be higher, skewing the result of the prevalence of the condition. However, the timing of data sampling has enabled us to make direct comparisons with the previously published LUNG-SAFE study, which used the same period [13]. As we have concentrated on the prevalence and initial treatment of AHRF, we have not collected detailed data on important treatment modalities, which might have affected outcomes such as tracheostomy, use of spontaneous breathing trials, rate of re-intubation, fluid management, daily interruption of sedation, use of thromboprophylaxis etc. Due to the nature of the study we also did not collect data on length of stay or longer-term outcomes. As the study sample was relatively small, we were only able to perform an univariate analysis to avoid potential overfitting of the model associated with a multivariate analysis. Nevertheless, our study has also proved that large-scale collaboration is possible in our hospitals and paved the way for other national studies [36,37]. As our study was concentrating on clinical variables, we did not collect detailed biomarker profiles for our patients and computer tomography was only performed in a quarter of the patients. Hence, detailed phenotyping of patients using these modalities, which has proved useful in the COVID-19 pandemic, could not be completed [38,39].

## 5. Conclusions

AHRF is associated with high mortality despite increased awareness and recognition of this condition. Despite advances in care, mechanical ventilation and use of adjunctive measures are still not optimal for the vast majority of patients.

## Figures and Tables

**Figure 1 jcm-09-03521-f001:**
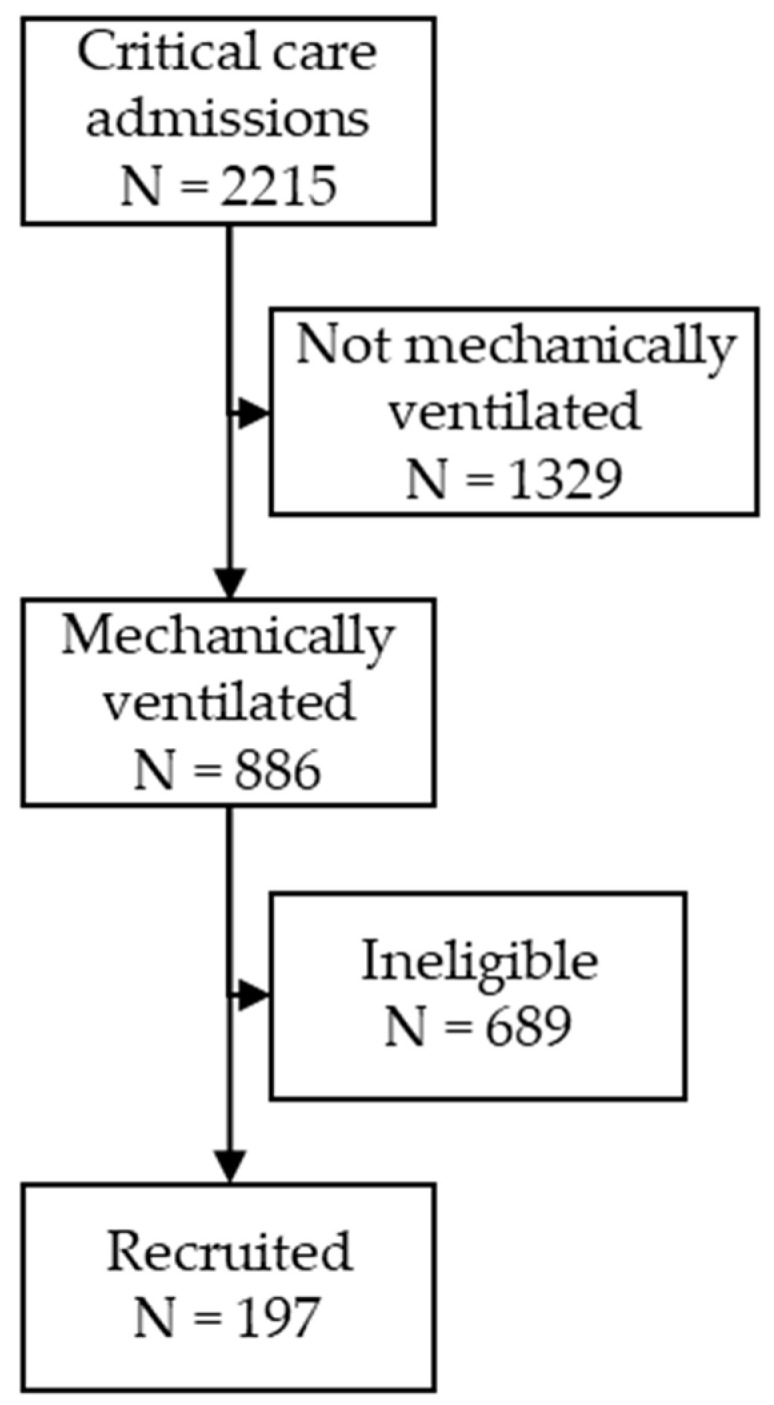
Prevalence and Outcome of Acute Hypoxemic Respiratory Failure in Wales (PANDORA-WALES) recruitment flow chart and eventual study sample. Mechanically ventilated patients who fulfilled study criteria of PaO2/FiO2 ≤ 300 mmHg on invasive mechanical ventilation with a Positive End-Expiratory Pressure (PEEP) of 5 cm H2O or more, and with a FiO2 of 0.3 or more were included in the study. All other patients were ineligible.

**Figure 2 jcm-09-03521-f002:**
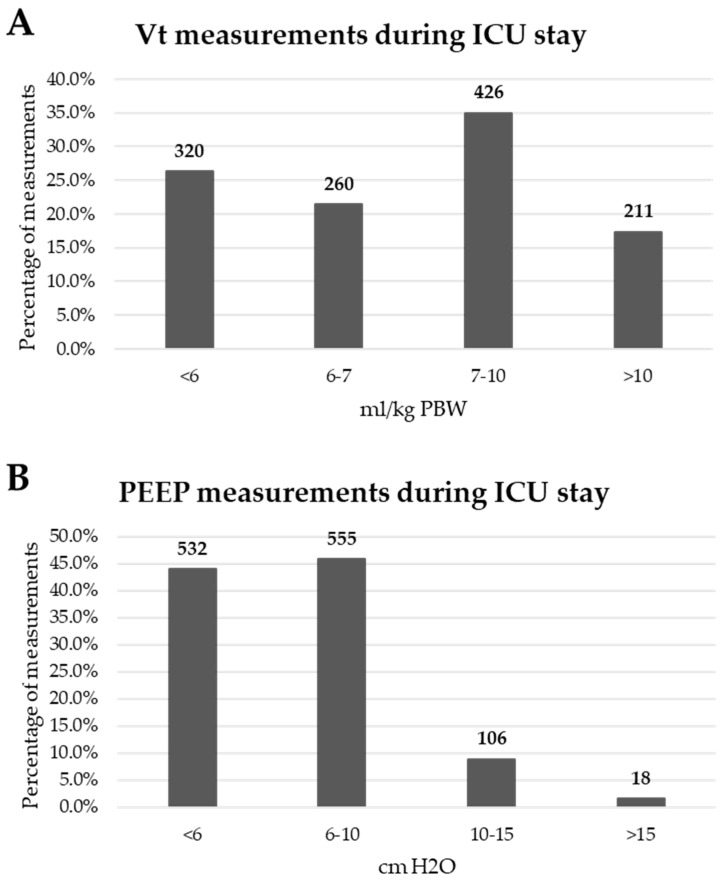
(**A**) V_T_ measurements during the whole ICU stay. (**B**) PEEP measurements during the whole ICU stay. Absolute numbers are shown above the bars.

**Table 1 jcm-09-03521-t001:** Summary of patient demographics and clinical characteristics.

Patient Characteristics	Number of Patients, Percentage (n = 197)
*Demographics*	
Age (years)	60 (49–70)
Male sex	117 (59.4%)
*Comorbidities*	
HTN	81 (41.1%)
HF	16 (8.1%)
Diabetes	27 (13.7%)
Obesity	26 (13.2%)
Liver cirrhosis	11 (5.6%)
Chronic kidney failure	17 (8.6%)
Immunosuppression	12 (6.1%)
Neuromuscular disease	1 (0.5%)
Malignancy	11 (5.6%)
Use of NIV at home	3 (1.5%)
Smoking	46 (23.4%)
Alcohol excess	32 (16.2%)
*Hospital type*	
Tertiary hospital	44 (22.3%)
District general hospital	153 (77.7%)
*Reason for ICU admission*	
Clinical condition	174 (88.3%)
Emergency surgery	32 (16.2%)
Planned surgery	2 (1.0%)
*Physiology at the time of recruitment*	
Baseline PaO_2_/FiO_2_–median mmHg (IQR)	150 (99–209)
V_T_–median mL/kg PBW (IQR)	6.9 (6.1–8.2)
Respiratory rate–median (IQR)	19 (16–23)
Plateau pressure–median cm H_2_O (IQR)	22 (17–25)
Peak pressure–median cm H_2_O (IQR)	23 (19–29)
PEEP–median cm H_2_O (IQR)	7 (5–10)
pH–median (IQR)	7.28 (7.17–7.37)
PaCO_2_–median kPa (IQR)	6.2 (5.2–7.9)

Data are median (IQR) or n (%). HTN, hypertension; HF, heart failure; NIV, non-invasive ventilation; VT, tidal volume; PBW, predicted body weight; PEEP, positive end-expiratory pressure.

**Table 2 jcm-09-03521-t002:** Impact of ventilation type on V_T_ classification.

	Ventilation Type		
V_T_	Volume Controlled (n = 126)	Pressure Controlled (n = 57)	*p* Value
V_T_ in optimal range	76 (60.3%)	25 (43.9%)	0.049
V_T_ above 7 mL/kg PBW	50 (39.7%)	32 (56.1%)	0.038

Ventilation types other than volume or pressure controlled were removed from the analysis due to small values. Optimal ventilation was described as less than 6 and between 6 and 7 mL/kg PBW. Data were missing for nine patients. *p*-value <0.05 is bold and underlined. For statistical analysis a Chi-square test was used.

**Table 3 jcm-09-03521-t003:** Intervention and monitoring modalities provided to patients throughout ICU stay.

Intervention and Monitoring	Number of Patients, Percentage (n = 197)
*During the ICU stay*	
Oesophageal pressure monitoring	1 (0.5%)
Pulmonary artery catheter	1 (0.5%)
Nitric oxide	3 (1.5%)
Vasopressor use	80 (40.6%)
Corticosteroids	56 (28.4%)
Systemic vasodilators	6 (3%)
Blood transfusion	12 (6.1%)
Renal replacement therapy	26 (13.2%)
High frequency ventilation	2 (1%)
ECMO	1 (0.5%)
ECCO2R	1 (0.5%)
NAVA	1 (0.5%)
PiCCO	4 (2%)
NIV after being extubated	1 (0.5%)
*On the day of recruitment*	
Neuromuscular blockade	42 (21.3%)
Recruitment manoeuvres	35 (17.8%)
*Prone by protocol*	
First 48 h	2 (1%)
During the evolution	1 (0.5%)
As a rescue	1 (0.5%)

Data are presented as n (%). ECMO, Extracorporeal membrane oxygenation; ECCO2R, Extracorporeal carbon dioxide removal; NAVA, Neurally Adjusted Ventilatory Assist; PiCCO, Pulse index Contour Cardiac Output; NIV, non-invasive ventilation.

**Table 4 jcm-09-03521-t004:** Complications developed by enrolled patients. Data are presented as n (%).

Complication	Number of Patients, Percentage (n = 197)
Pneumothorax	2 (1%)
Pneumonia	36 (18.3%)
Pleural effusion	11 (5.5%)
Atelectasis	4 (2%)
New lung infiltrates	21 (10.7%)
Pulmonary oedema	5 (2.5%)
Shock	27 (13.7%)
Sepsis	38 (19.3%)
Major surgical operation	16 (8.1%)

**Table 5 jcm-09-03521-t005:** Univariate analysis of factors associated with patient mortality. Data are median (IQR) or n (%). *p*-value < 0.05 is bold and underlined.

Risk Factor	Survivor (n = 125)	Non-Survivor (n = 70)	*p* Value
Age	57 (47–68)	65 (52–72)	0.034
SOFA score	8 (6–9)	9 (7–11)	0.019
Alcohol excess	15 (12.0%)	17 (24.3%)	0.026
Inotropic support	73 (58.4%)	51 (72.9%)	0.044
Recruitment manoeuvres	17 (13.6%)	17 (24.3%)	0.059

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
