# Peer review of "Prevalence and Outcomes of Acute Hypoxaemic Respiratory Failure in Wales: The PANDORA-WALES Study"

_jcm, 2020, doi:10.3390/jcm9113521_

Round 1

Reviewer 1 Report

The authors conducted a multi-center, prospective, observational study in Wales (UK) namely PANDORA-WALES with the primary intent to assess the mortality rate in the ICU for patient suffering for AHRF. Secondary end-point was to investigate the values of parameters of ventilatory management as well as the use of rescue therapies for hypoxemia.

While i pay my deepest compliments to the authors for their work i think that some issues must be settled in order to publish this manuscript.

  • In the abstract section the acronym ARDS should be written in full
  • Authors state : " The diagnosis of respiratory failure was performed using chest X-ray in the majority of the 110 patients (86.3%, 170/197). Other modalities of diagnosis included CT scan (24.4%, 48/197); blood 111 culture (19.8%, 39/197); sputum culture (18.8%, 37/197); echocardiography (14.2%, 28/197); 112 bronchoscopy (8.6%, 17/197) and bronchoalveolar lavage (4.1%, 8/197). Lung biopsy and autopsy 113 post-mortem was not performed in any of the cases." I think this sentence is incorrect. The diagnosis of respiratiry failure requires an ABG. 

Reviewer 2 Report

GENERAL COMMENTS

The authors have assessed the incidence and outcomes of acute hypoxemic respiratory failure (AHRF) in mechanically-ventilated patients of 9 ICUs of Wales. They report that 22% ventilated patients had AHRF at admission, and they had 35% ICU mortality rate. The use of several therapies is also described, as well as a univariate analysis of variables associated with mortality.

I have some comments and queries that may help improving the manuscript.

SPECIFIC COMMENTS

Major comments

  1. Article title. Rather than prevalence, which is commonly used as the frequency of chronic diseases in a given population, this study may be better named as “Incidence and outcomes of AHRF in mechanically ventilated ICU patients of Wales”, or similar. AHRF is by definition an acute, not a chronic disease, and a representative sample of the Welsh population was not selected. This applies to other parts of the manuscript where the authors refer to prevalence.
  2. Causes of AHRF. AHRF is an “umbrella” under which many different clinical conditions are included. The different causes of AHRF are relevant information that should be reported in the manuscript.

Minor comments

  1. Recruitment period. An explanation of the reason for choosing two 2-month periods is advisable. Maybe autumn and winter may result in different proportion of AHRF than spring or summer, and therefore the reported incidence of AHRF may not be completely accurate.
  2. Introduction, page 2, second paragraph. The primary and secondary end-points reported here are not consistent with those reported in the abstract. This need to be clarified.
  3. This section is very short and the authors have not clearly reported what they compared (i.e., it seems that the compared survivors and non-survivors). This needs to be specified.
  4. Figure 1. The reasons for ineligibility should be described. In addition, were all mechanically ventilated patients registered?, or those with a minimal duration of ventilation? Please, explain.
  5. Table 1, legend. The diagnosis of respiratory failure is done by arterial blood gases. What the authors report in this legend is the number and proportion of diagnostic examinations done in their population. This needs to be clarified.
  6. Results, page 4, subheading 3.2, second line. “PEEP of 5 or less”, instead of “at least 5”? Moreover, are these percentages of PEEP values of patients or measurements? Please, clarify this issue.
  7. Figure A. Both plots would be more informative if the Y axes reflect the % of measurements, and if the absolute number is shown above the bars.
  8. Results, page 5, first paragraph. What do the authors mean with “optimal Vt? In the discussion section, it seems that this is 7 or less mL/kg IBW, but this should be explained in the results.
  9. Results, page 7, first paragraph. It’s not clear if the incidence of ARDS is cases at admission or cases developed during the ICU stay. the median value of 0 days could be that they are mixing cases with ARDS at admission with those that developed during their ICU stay. A clear definition of this issue is needed.
  10. Table 5, and discussion. The univariate nature of the analysis of mortality without any adjustment of confounders has limited value. This should be added to the limitations section at the end of the discussion.

Round 2

Reviewer 1 Report

Authors fixed all mentioned issues.

Author Response

Many thanks for the comments.